# Financial Behavioral Health and Investment Risk Willingness: Implications for the Racial Wealth Gap

**DOI:** 10.3390/ijerph20105835

**Published:** 2023-05-16

**Authors:** Jeffrey Anvari-Clark, Theda Rose

**Affiliations:** 1Department of Social Work, University of North Dakota, Grand Forks, ND 58202, USA; 2School of Social Work, University of Maryland, Baltimore, MD 21201, USA

**Keywords:** financial behavioral health, investment risk willingness, financial well-being, financial precarity, financial self-efficacy, financial instability

## Abstract

Financial behavioral health (FBH) influences numerous socio-ecological domains, affecting investment risk willingness and consequent wealth levels. The experience of FBH by racial group is unknown, and findings of differences between Black and White investors’ risk willingness are mixed. The study’s aims are to establish an FBH measure and explore its application to risk willingness by racial group. The study used a subset of data from FINRA’s 2018 National Financial Capability Study, including Black (*n* = 2835) and White (*n* = 21,289) respondents. Through factor analysis, 19 items were confirmed for the FBH measure; the measure was then applied to investment risk willingness using structural equation modeling (SEM). Invariance analyses showed that the FBH model had an excellent fit for White respondents but not Black respondents. The SEM analysis determined that FBH accounted for 37% of the variance in risk willingness (*R*^2^ = 0.368; β = 0.256, *p* < 0.001). Racial group affiliation was a negligible predictor of risk willingness (β = −0.084, *p* < 0.001). This project contributes an empirical basis for FBH, emphasizes the importance of FBH for investment risk willingness, and elucidates that racial group differences in risk willingness could be an unlikely contributor to the wealth gap.

## 1. Introduction

A person’s thoughts, feelings, and behaviors toward money can have a meaningful impact on both their financial state and behavioral health [1]. Relationships between particular aspects of financial state and behavioral health (such as between debt and mental and physical health) have been sufficiently studied [2], yet no literature until now has broadly explored the psycho-behavioral relationship between finances and behavioral health or looked at how this relationship impacts subsequent decision-making, such as investment risk willingness. This study explores the concept of financial behavioral health (FBH), which is comprised of financial precarity (FP), financial self-efficacy (FSE), and financial well-being (FWB), how the FBH model varies by racial group, and if the influence of FBH on willingness to take risk when investing is moderated by race. For this paper, race is defined as a social construct of hierarchical classification based upon skin color. The implications of presumed superiority manifest themselves as racism, in which differential treatment is imposed. The current research expands our understanding of the financial domain as not just a social determinant of health, as it is frequently conceptualized [3,4], but to appreciate its psycho-behavioral impact and to introduce racial group affinity as a potential moderator of the effects of low FBH. Such insights could then have meaningful impacts on macro-level policies and programs, mezzo-level work by behavioral health practitioners, and at the micro level as individuals build resilience, in the presence of racism and challenging financial predicaments. Further implications for understanding and addressing the impact of these differences on perpetuating the racial wealth gap, straining family networks, and improving FBH are discussed.

## 2. Financial Behavioral Health (FBH)

In this paper, behavioral health is considered the domain of interinfluencing components that determine health and well-being due to patterns of thought and action. Depending on the definition, several of the following components are often included [4,5]: mental health, physical health, coping health, and social health. However, the present conception of FBH is relatively new, with little precedent. In their article measuring the relationship between health insurance and economic behaviors, Mitchel and Shan [6] use the phrase “financial behavioral health” once in terms of measuring key behaviors, such as saving for emergencies or retirement or using certain types of loan products. Anvari-Clark referenced the phrase in a blog post article exploring financially related stress but did not define it [7]. Finally, Wolfsohn used the phrase in two webinars, in which she provided a transtheoretical model adapted to financial behaviors [8], and then provided a pathologically oriented definition of FBH as “identifying, addressing, and reducing the problems, stress, anxiety, shame, and other harmful feelings that result when money, physical, mental, and emotional health negatively converge” [9]. No further discussion of FBH was offered. Thus far, to our knowledge, no theoretical exploration of FBH has been conducted to explicitly situate it within a larger behavioral health framework, nor has any empirical research studied FBH as a concept unto itself.

The perception of financial experiences, money-related habits and attitudes, and the resulting sense of security, or lack thereof, are subjective and positioned according to one’s internal frames of reference. One’s relationship with money responds, at an individual and family level, to behavioral interventions (such as financial coaching or cognitive behavioral therapy) [10]. As these subjective financial elements intertwine with the other behavioral health components and likewise contribute to health and well-being, it is proposed that FBH be included as an additional domain to those that are considered to comprise behavioral health. Initially, elements of FBH could include FP (a state in which both material and psychological well-being are impacted by negative financial conditions and perceptions), FSE (the ability and confidence to make financial decisions and perform financial tasks) [11], and FWB (the measure of one’s sense of security and freedom of choice, both in the present and future) [12].

Conceivably, when there is higher FP and lower levels of FSE and FWB, one may have higher levels of financially related anxiety and distress (such as due to an inability to meet multiple demanding financial obligations). Feelings of hopelessness, worry, and powerlessness to control the situation or see options for improvement may manifest. For people who experience racism and discrimination, a lower FBH because of their lived experiences may impact the nature of their financial decisions. Ultimately, these decisions then influence a trajectory for health and wealth, the latter of which is the primary focus of the study.

It should be noted that much of the literature on financial concerns and behavioral health focuses on the experiences of predominantly White sample groups, unless specified otherwise. In most cases, the use of a race variable is simply noted as a demographic descriptor, but discussion of the implications of racism on the given topic is absent. The focus of this study is on the implications of structural racism as enacted in policies, laws, and norms in institutions and society, particularly as it manifests in the design of research, products, and tools, in differential access to income, wealth, and other resources, and results in differences in socioeconomic status (SES) and FBH, even after controlling for other factors. Following is an overview of the three components of FBH—FP, FSE, and FWB—in which each is explored to understand their contribution to FBH.

### 2.1. Financial Precarity (FP)

FP refers to a state in which both material and psychological well-being are impacted by negative financial conditions and perceptions. However, in the extant literature, there is often a focus on either the material or the psychological. Meuris and Leana [13,14] refer to it simply as “worry about one’s financial situation”. Gaffney et al. [15] frame FP as being unable to cover an unexpected $400 expense without selling possessions or going into debt. The literature on scarcity [16,17] from the fields of behavioral economics and behavioral psychology highlights the impact of worry due to material circumstances, thus helping to understand the connection between the material and psychological aspects of FP. This then warrants a definition that encapsulates both aspects. FP is often conceptualized similarly to financial insecurity, although the term insecurity could refer to a future state of financial difficulty in the event of a financial shock. FP may then be differentiated as a present state of financial difficulty and its concordant negatively impacted emotional or psychological state.

Broadly, FP can negatively impact home and family life as well as work life [13,14]. It stems from insufficient income or wealth to meet basic needs, thereby being most prevalent among individuals and families with traditionally measured low-income. However, it is also frequently found among those characterized as Asset Limited, Income Constrained, Employed (A.L.I.C.E.) [18], who are typically working- and middle-class.

Significant research has explored FP and financial insecurity. Beverly’s work on material hardship [19] introduced the importance of objective indicators to measure difficulties accessing various goods and services (beyond simply assessing differential income levels). Subsequent research has focused on the impact of structures, environments, and poverty on a person’s financial state. Insufficient and volatile income [20,21,22]; inadequate access to affordable and appropriate banking services [23,24]; overwhelming medical expenditures and debt due to poor insurance coverage [25]; lack of affordable housing, historic and current redlining, racially biased property appraisals, and discriminatory mortgage lending [26,27,28]; and more recently, the dramatic increase in the cost of higher education and the need for student loans amidst questionable returns [29,30]; these and other environmental and structural issues make life both more expensive and more difficult to manage, with fewer resources to do so, than it is for those who have more income and wealth at their disposal. Yet, important as these objective measures of material hardship and structural issues are, they alone do not capture the impact of subjective perceptions, such as worry, concern, and a sense of control, that also contribute to the proposed definition of FP. 

Finally, under the financial capability paradigm, it is presumed that if one has the ability and opportunity to act, they will consistently carry out choices that will improve FWB [31]. However, this may not account for the perception of scarcity, which can subconsciously sabotage one’s behaviors and capacities to meet immediate needs at the expense of longer-term outcomes [16]. Thus, understanding the role FP has in behavioral health requires appreciation for varying co-influencing subjective and cultural perspectives. 

### 2.2. Financial Self-Efficacy (FSE)

FSE is the ability and confidence to make financial decisions and perform financial tasks [11,32]. In some literature, the terms economic self-efficacy and financial self-efficacy (FSE) are used interchangeably [11]. However, economic self-efficacy can also refer to a more outward, macro-oriented ability to adapt to changing economic conditions [32], such as the ability to learn more marketable skills to obtain a better job. While there is still some overlap conceptually and in their use in the literature, this distinction can set FSE apart for the purpose of this paper as being more inward-oriented, where the context is generally limited to proficiency with financial matters within the household.

A key aspect of self-efficacy generally and FSE specifically pertains to the importance of personal agency and the belief that one can set, pursue, and achieve meaningful and worthwhile goals [32,33,34]. There is a perception of confidence in one’s capability to navigate financial concerns, and it can be considered a building block to economic empowerment [35]. Higher levels of FSE could imply proactive management of personal financial affairs, whereas lower FSE may relate to worry, procrastination, ignoring, abdicating responsibility for handling financial affairs, or generally finding it difficult to navigate financial challenges, be they chronic (such as income volatility) or acute (a financial shock). 

What may be illustrative of FSE is the type and degree to which one uses tools to manage finances: analog or digital tools, computer programs, websites, or apps. These could include a budget to proactively allocate money or a spending tracker to reactively monitor expenses, credit monitoring apps to track changes in one’s credit report and/or score, investment apps for growing wealth, and digital banking tools to monitor account balances, make transfers, or employ automatic bill pay features. An aspect of the financial capability model [31] emphasizes product and service accessibility, which could be driven by racialized marketing. However, the decision to use any of these and how could reflect one’s desire to improve their financial behaviors, their comfort handling money, and the degrees of risk and loss aversion they may hold. Higher FSE may be correlated with the perception of having “enough” money in one’s account that there is little worry they will overdraw accidentally during an auto bill pay occurrence. Lower FSE may manifest in the conscious or unconscious decision not to budget or monitor spending for fear that facing the true state of their financial decisions and circumstances will force them to confront cognitive dissonance between the way they want to see themselves and the true consequences of their actions.

According to Bandura [33], general self-efficacy may not carry over between life domains; a person may have higher self-efficacy in their career but lower levels in their role as a parent. However, O’Neill et al. [36] found a positive relationship between budgeting and engaging in health-promoting activities, such as eating well and exercising. (It was unclear if this carried across different demographic groups as most respondents were White females with higher levels of education.) It would seem then that there could be some multidisciplinary application of meta-learning and behavioral tools, such as the use of accompaniment, setting S.M.A.R.T. financial goals [37], and employing grand gestures (making a significant investment to achieve a desired outcome), and that once a person has successfully used these in one area of their life, they may be more apt to employ such behavior modification tools in other life domains. There could also be an overlap in benefit; using and sticking to a grocery shopping list could be used to improve and manage life in both the domains of physical health and nutrition and to mitigate impulsive spending.

### 2.3. Financial Well-Being (FWB)

Depending on how it is addressed and the resources utilized, FP can result in a positive or negative perception of FWB. Different definitions of FWB exist [38]; the most accepted and utilized is by the U.S. Consumer Financial Protection Bureau (CFPB) [12], defined as a subjective assessment of one’s sense of financial security and choice, both currently and in the future. Netemeyer et al. [39] further explicate this concept into two distinct domains: current money management stress and expected future financial security. Both conceptualizations utilize a series of robust questions to produce an index score. However, without these questions included in a given data set, proxy questions relating to satisfaction with one’s financial state can approximate similar insight [40], where the focus is on a subjective perception rather than an objective measure, such as a credit score or income. 

As both the CFPB [12] and Netemeyer et al. [39] research show, FWB is a multifaceted concept. That stated, higher FWB has been found to be associated with both larger amounts in savings as well as stronger savings behaviors [41], a form of FSE. FWB subsequently contributes substantially to overall perceived well-being [39], which could then suggest its importance for behavioral health.

### 2.4. Conceptual Model

A model applying the theoretical relationships that characterize FBH to outcome financial decisions (and for the purpose of this study, investment risk willingness specifically) is shown in Figure 1.

The domains of behavioral health impact each other. As such, the financial domain of behavioral health may impact one’s mental, physical, coping, and social health. In Figure 1, FBH is explicated as the resulting outcome of FP, FSE, and FWB. However, it is also conceivable that one’s existing FBH conversely impacts levels of FP; higher FBH would lead to lower levels of FP through the ability to mitigate and handle financial crises. If one finds themselves in a difficult financial position, they could deal with the precarity by employing strategies to handle the situation (FSE). If not, or if they have lower levels of FSE, then it results in lower levels of FWB and thereby lower FBH, resulting in poorer mental health, manifesting, for instance, as anxiety or an adverse spending habit. Depending on the strengths of the individual, their resources, and their coping strategies, the cycle can become self-reinforcing in either positive or negative trajectories. Together, the state of FBH affects the intentions of the individual in terms of how they will approach making financial decisions generally and, in this study, their willingness to take on investment risk specifically. 

The second half of the model illustrates how people may navigate financial decisions. The presence of a scarcity mindset or outlook, indicated by low FBH, could then influence investment decisions. However, cultural expectations and patterns of financial socialization could also shape the extent to which a person makes financial decisions based upon independence values (where the focus is on the financial security of oneself and perhaps a nuclear family) or also interdependence values, in which expectations to share wealth with extended family, friends, or neighbors are much greater. Although these approaches are not exclusive to one cultural group, the expectation to share resources generally differs between Black Americans and White Americans [42]; Black people generally have greater expectations of ongoing interdependence with kin both within and outside the nuclear family [43]. Sharing among Black, Latinx, Asian, and other racial/ethnic marginalized groups, particularly at lower income levels, has garnered some attention from scholars [44,45,46,47], but more remains to be explored. For example, those experiencing racism and scarcity may, accounting for more financially interdependent oriented values, have a lower willingness to take on investment risk than others also having lower FBH. (Conversely, initial research on “cushioning” would suggest a higher willingness to take risks [48].) Recognizing that other kin and extended family members will be impacted by one’s own financial state (and the expectation to support them, especially in times of hardship), minimizing the risk of loss through investment could be a method of hedging one’s wealth to ensure its presence when needed by others. This could be one explanation for research findings that Black Americans typically have lower investment risk tolerance than White Americans [49,50].

A recent study by Hasler et al. [51] found that significant debt, low financial reserves, and difficulties managing money were contributors to higher levels of financial anxiety and financial stress. Another study found connections between higher anxiety and lower investment risk tolerance [52]. Accordingly, should the FBH of the individual be worse as an outcome of the broader constructs of higher FP and lower FSE and FWB, it would be expected that this would also lower one’s risk tolerance. Black Americans have slightly lower raw FWB scores (*M* = 52) compared with White Americans (*M* = 56, scale range 19 to 90; CFPB, 2017), although the wide range of scores may suggest little significance in the difference between groups. Notably, when accounting for covariates, Lusardi found higher FWB scores among African American and Hispanic Millennials than White Millennials [53]. Furthermore, it has not been assessed with this same data set if Black Americans were also more likely than White Americans to experience financial shocks, including providing unexpected financial support to a family member or friend. 

What research does exist on financial interdependence at a familial level (whether conceptualized as nuclear or more broadly to include relatives, kin, and close relations) is mixed and often depends upon class considerations [42,43], and the definition of family [54,55]. How this translates into better or worse FBH in the presence of scarcity on the part of those who are sharing and those who receive, however, is not clear, nor is it understood how this impacts financial choices.

## 3. Research Aims

The first stage of this study will explore the components of FBH—FP, FSE, and FWB—and their relationships with FBH. This will entail constructing component parts of the FBH model (Aim 1) and then measuring the full FBH model (Aim 2). Measuring the full FBH model will help establish an initial empirical basis for how the various financial indicators contribute to FBH as a conceptual domain. Due to limitations in the data set, the full modeled relationship between FBH and the mental, physical, coping, and social aspects of behavioral health cannot be assessed; this will be an area for future research. 

Next, due to racism specifically, Black individuals and families face a host of challenges within multiple domains of their lives, above and beyond those that might occur by chance. According to stress and life course theory, people with the accumulation of these challenges would then present with poorer behavioral health generally and worse FBH specifically than they would without the added traumas, discriminations, financial barriers, and stressors in their lives. The study will then explore differences in the fit of the FBH model according to racial affinity group, with a focus on differences between survey respondents who self-selected Black and/or White (Aim 3).

It should be noted that recent scholarship in health sciences research has advocated for a more technically accurate and socially conscious use of the race variable [56,57]. On its own, the race variable has been used as a proxy to represent biological difference, just as income levels are often used as a proxy for social class. This use of the race variable stems from its origins as a social construct to justify the oppression of Black people. Although these supposed biological differences have been thoroughly found to be false, the social impact of the myth is real [58]. The differences between people based on racial category should instead be interpreted as the consequence of racism and the effects of inequitable political, economic, and social treatment [56]. Editorial [59,60] and theoretical [56] scholarship has called for reframing the use of the race variable, with Phelan and Link [61] illustrating how racism itself has a fundamental causal relationship with health, separate from any effect SES may also have. To capture this difference methodologically, variables that measure levels of discrimination based on racism, controlling for SES measures, are ideal [61]. However, in using the race variable itself to provide insight into differences caused by racism, new literature is providing some guidance. Principally, definitions, interpretations, and discussions of race should be set in the sociopolitical context (not biological), for which race serves to represent differences in social experiences [62,63]. Assari and Curry [64] provide an example of this in their empirical study, using the race variable as a social determinant of a health outcome (racial difference in child thalamus volume), representing the impact of disparate access to resources and social inequality. For the present study, the use of the race variable represents the individual’s relationship with society generally [58] and the research questions specifically, particularly in terms of how structural racism deprives Black Americans of financial needs and opportunities, deprivation being a form of scarcity. (In this regard, it should be noted that access to investment products has traditionally been dependent upon both having excess money to invest and access to the products. Due to differentials in pay, wealth extraction tactics, and exclusion from banking and other wealth-building institutions, it has historically been difficult for Black Americans and other minority groups to make such purchases.) Findings of difference by racial group may thus be interpreted to reflect the impact of structural racism on the conceptualization of FBH, a person’s resulting FBH outcome, and investment risk willingness.

Finally, in Aim 4, the authors explored how FBH influences financial decision-making (specifically in terms of willingness to take on investment risk), as moderated by race. It was hypothesized that financial interdependence as a cultural value and a resilience mechanism (as assessed by the race variable) moderates the outcomes. A person’s FBH can have a profound impact on their life and influence how financial decisions are made. Preliminary research has suggested that heightened emotional states, such as anger and anxiety, can impact investment decisions [52,65], and a review of investing literature published between 1960 and 2014 found that being happy was affiliated with greater risk tolerance [50]. Additionally, when scarcity is present in a person’s life, it “captures” their attention to focus on only the most urgent of needs, often causing them to make decisions and choices that ignore longer-term consequences [17]. (It should be emphasized that the inclusion of scarcity-as-racism in this context is not a comparison between people—as is the typical use of a race variable—but between conditions and how a person might behave in different circumstances.) For the reason that investment risk willingness has been shown to impact equity allocation, which influences wealth accumulation [50,66], assessing FBH’s contribution to a person’s risk willingness may serve as a helpful application of the model with potentially consequential implications. Findings of differences between Black and White investors’ risk willingness have been mixed [50,65,67,68], and this study will help contribute to the existing literature.

## 4. Materials and Methods

### 4.1. Sample and Source of Information

The study used extant data from a disaggregated version of the U.S. based 2018 National Financial Capability Study (NFCS; State-by-State data set), a project funded by the FINRA Investor Education Foundation [69]. NFCS is a nationally representative study with waves conducted in 2009, 2012, 2015, and 2018. Only the 2018 wave included questions on financial anxiety or financial stress. Given the importance of these mental health measures to the present study, data sourcing was limited to this fourth wave. 

Approximately 500 adults (18+) in each state across the U.S., plus Washington, DC, were included in the study [70]. The total sample size was *N* = 27,091. Oversampling in two states provided 1250 respondents. The survey was administered to respondents online using non-probability quota sampling from panels provided by SSI (Survey Sampling International), EMI Online Research Solutions, and Research Now. Nominal incentives were offered in exchange for participation. Quotas for each state were set by age for gender, ethnicity, education level, and income level based upon census track data. 

For the survey, a total of 1,410,923 email invitations were sent to potential respondents on the panels, for a total of *N* = 27,091 respondents in the data set. The full sample (*N* = 27,091) was used to establish the FP and FSE constructs. To assess the FBH model and its application, the study utilized a subset of responses from those who identified as Black (*n* = 2835) or White (*n* = 21,289) according to their racial group affiliation, which further reduced the sample size (*n* = 24,124). 

### 4.2. Measures

#### 4.2.1. Aim 1: FBH Component Development

A measure of FP was constructed, employing exploratory factor analysis (EFA) and then confirmatory factor analysis (CFA), to assess material and psychological well-being as impacted by negative financial conditions and perceptions. The measure assessed 16 potential items gathered from the data set. Items are listed in Table 1. The objective and subjective measures were intended to reflect material (objective) and psychological (subjective) aspects of precarity and were initially positioned to be consistent with prior findings [15,71]. Several of the questions included in the FP measure have also been used to measure FWB [71]; in this case, their usefulness in measuring precarity, in combination with additional components, was assessed. 

FSE was also assessed using an EFA to CFA approach with the available measures from the data set that reflected concepts associated with existing measures of this construct [11,32]. FSE was expected to be positively related to the respondents’ rating of FWB (as financial satisfaction). See Table 1 for the list of FSE variables assessed.

Finally, financial satisfaction, as a proxy for FWB, was measured on a scale of 1 = not at all satisfied to 10 = extremely satisfied in response to the question J1 (29) “Overall, thinking of your assets, debts, and savings, how satisfied are you with your current personal financial condition?” 

#### 4.2.2. Aim 2: FBH Construct

In order to explore the relationships that the FBH components have with FBH while accounting for financial anxiety and financial stress, variables were used according to the model illustrated in Figure 2.

Outcome Variable. FBH was the main construct, as a second-order latent variable. Per the definition used in this study, FBH entailed the contributions of FP, FSE, and FWB (using financial satisfaction as a proxy). Financial anxiety and financial stress were initially modeled with regression as directly influencing FBH. This was in alignment with recent findings by scholars using the same NFCS data set in which certain items from the FP and FSE concepts were associated with financial anxiety and financial stress [51]. Financial anxiety was measured using response scores to the statement “thinking about my personal finances can make me feel anxious” that ranged from 1—strongly disagree to 10—strongly agree. Financial stress was measured using response scores to the statement “discussing my finances can make my heart race or make me feel stressed” that ranged from 1—strongly disagree to 10—strongly agree. In both cases, higher scores suggested higher anxiety and stress levels. 

Indicator Variables. Three measures comprised the indicators of the FBH latent variable, drawing upon the suite of respective observed independent variables taken from the data set. FP and FSE were developed according to an EFA/CFA treatment. Greater FP was anticipated to be negatively associated with decreased FBH. Increased FSE would be positively associated with increased FBH. Since many of the questions used for assessing FWB were already utilized to measure FP in this study, the proxy variable—financial satisfaction—was used, assuming a high correlation with the other FWB measures. 

Control Variables. Other demographics have been recognized as important contributors to financial state, including gender [72,73], age [40], and income levels [39]. Gender was measured as a binary variable, coded as 1 = male, 2 = female. Age was measured on an ordinal scale of grouped ranges, coded as 1 = 18–24, 2 = 25–34, 3 = 35–44, 4 = 45–54, 5 = 55–64, and 6 = 65+. Income was also measured on an ordinal scale of grouped ranges, coded as 1 = Less than $15,000, 2 = At least $15,000 but less than $25,000, 3 = At least $25,000 but less than $35,000, 4 = At least $35,000 but less than $50,000, 5 = At least $50,000 but less than $75,000, 6 = At least $75,000 but less than $100,000, 7 = At least $100,000 but less than $150,000, and 8 = $150,000 or more.

#### 4.2.3. Aim 3: FBH Differences by Racial Group Affiliation

In order to assess the differences in the fit of the FBH model by racial group (i.e., assessing model invariance), a subgroup with two categories (one for Black respondents and one for White respondents) was created. Findings were interpreted to explore the impact of structural racism in model design [56,59,60,61,62,64]. The variable constructed for this study listed respondent racial identity affiliation as 1 = Black and 0 = White.

#### 4.2.4. Aim 4: FBH on Investment Risk Willingness

Finally, it was important to determine how FBH and racial group affiliation (with their represented differences in financial values and racism-based experiences) may influence financial decision-making. Willingness to take on investment risk was measured using a single variable on a scale of 1 to 10, with higher numbers representing increased risk willingness. The measures for FBH served as predictor variables, and the risk willingness measure served as the outcome variable. The thought processes required for making sound investment decisions require a high degree of executive functioning. The variable related to taking investment risk therefore seemed to encapsulate this mental exercise. The hypothesized relationship between FBH and risk willingness is portrayed in Figure 2.

### 4.3. Data Analyses

A full group sample was used (*N* = 27,091) to construct the FP and FSE measures as components of FBH (Aim 1). Additionally, a two-group racial category (*n* = 24,124) was compiled from the full sample for the construction of the FBH model and assessment tests of comparison and invariance (Aims 2 and 3). The two-group data subset was subsequently used for the application of the FBH model on investment risk willingness (Aim 4). Groups were stratified by racial category among White and Black respondents in terms of individualist and individualist/collectivist approaches to handling finances, combined with potential exposure to racism-based scarcity. Although testing for reasons why any differences exist was not conducted, assessing if racial group affiliation dictates any portion of the variance in risk willingness helps clarify mixed findings in the literature. The White respondents’ group (*n* = 21,289) was comprised of individuals who indicated their racial and ethnic identities were only White and non-Hispanic. The Black respondents’ group (*n* = 2835) was comprised of individuals who indicated their racial identity as being Black or African American, including those who may have selected additional racial or ethnic categories. Univariate and stratified analyses were conducted to understand the full sample population as well as the demographics and differences (*t*-test and Chi-Square test) between the constructed groups. Samples were assessed for missingness (≤5%), skew (|<3.0|), and kurtosis (|<10.0|) and found to be within acceptable levels for all [74]. Due to the large sample size and random low missingness rates (assessed through univariate analysis and review of question conditionality), listwise deletion was applied [75]. 

A structural equation modeling (SEM) approach using the R statistical programming language [76] with the lavaan package [77] was employed to address the study’s aims. The analysis progressed according to a jigsaw piecewise modeling technique [75,78], whereby components were fit individually and then sequentially added together while ensuring continued fit through the process. Covariates were only then added to the final model, according to theoretical and empirically based rationales and an improvement in ultimate model fit. This process included establishing FP and FSE latent variables through EFA and then CFA approaches. Once the relevant items and arrangements for each latent variable achieved appropriate fits (Aim 1), they were then modeled together with FBH as a second-order latent variable (Aim 2). Next, differences in model fit by racial group were assessed using measurement invariance testing (MI; Aim 3). Finally, after achieving an appropriate fit of the full FBH model, the variance contribution of FBH as regressed on investment risk tolerance was assessed using a general SEM approach with control variables (Aim 4). 

The methods used for the study were equivalent to a graded response item response theory (IRT) model, particularly for exploring the FP measure [79,80]. However, given the context of the larger project, an EFA/CFA/SEM parameterization was adopted to confirm factor loadings and structures and then provide a more congruent sequence of analyses for exploring the full theoretical model and its application. The preliminary applied model, as hypothesized, is represented in Figure 2.

To carry out EFA, oblique rotations have been recommended for cases in which many of the factors are correlated [81]. Geomin and oblimin rotations, both appropriate for use with this study [82], were tested; both provided similar fit values, but the traditional oblimin rotation presented more robust z-scores and corresponding p-values and was thus used for the final EFA models.

The EFA, CFA, and SEM analyses all utilized the WLSMV estimator (following a test of other estimators). This uses diagonally weighted least squares (DWLS), combined with mean- and variance-adjusted (MV) test statistics [83]; it is the lavaan default and generally recommended estimator for ordinal data [84]. Since the WLSMV estimator was used, survey weights were not added to the model (use of weight matrices is part of the calculation) [85]. To simplify modeling and analysis, all variable items were rendered ordinal, and in the SEM analysis, the function to standardize latent variables was used, fixing them at 1.000. Acceptable fits were assessed according to the following criteria: Standardized Root Mean Square Residual (SRMR) < 0.05, comparative fit index (CFI) and Tucker-Lewis index (TLI) > 0.95, and root-mean-square error of approximation (RMSEA) between 0.05 and 0.08 indicating sufficient fit, with an upper bound 90% CI < 0.08 [75,86]. For EFA, the Kaiser-Meyer-Olkin (KMO, desired level ≥ 0.70) [87] was used as a preliminary measure of factor adequacy; a minimum factor loading threshold of >|0.40|was used to determine which items to retain in the model construction [74,88].

## 5. Results

### 5.1. Sample Description

Of the full sample (*N* = 27,091), 56% identified as female and 44% identified as male. Those who identified as non-Hispanic categorized themselves as White (74%), Black (10%), Asian (4%), and Other (3%), whereas those who identified as Hispanic comprised 9%. The mean age of the full group was 48 years (*SD* = 17 years). Most respondents (73%) had at least some college education or more. Respondents reported a normal distribution of annual household income levels. Approximately one-third (35%) of respondents indicated having financially dependent children. The mean score for satisfaction with one’s finances (range from 1 = not at all satisfied to 10 = extremely satisfied) was *M* = 5.72 (*SD* = 2.88). Financial anxiety and financial stress scores were similar (scale ranges 1 to 7, with higher scores indicating higher levels of financial anxiety or stress), with *M* = 4.51 (*SD* = 2.02) and *M* = 4.12 (*SD* = 2.07), respectively. When considering willingness to take investment risks (ranging from 1 to 10, with higher scores indicating increasing willingness), respondents reported an average score of *M* = 4.91 (*SD* = 2.68). Table 2 presents the descriptive characteristics for the full sample in addition to the two constructed racial groups. The correlation matrix (Appendix A, Figure A1) shows the bivariate relationships of all variables according to Spearman’s rank correlation.

There were notable differences between Black and White respondents in every category except gender. On average, Black respondents were younger, had slightly lower education levels, lower household income, more financially dependent children, moderately lower financial satisfaction, and slightly higher levels of financial anxiety and financial stress than did White respondents. Despite these more adverse scores, the Black respondents also notably had a higher average investment risk willingness score (*M* = 5.361, *SD* = 3.071) than the White respondents (*M* = 4.788, *SD* = 2.613; *t* = 9.282, *df* = 3249, *p* < 0.001, 95% CI [0.452, 0.693]. Apart from gender, all tested comparisons were statistically significant to the level of *p* < 0.001. See Appendix B, Table A1, for group differences.

### 5.2. FBH Model Construction

The fit statistics of the study’s measurement models are shown in Table 3. The CFA and SEM analyses relied upon likelihood ratio testing (LRT) [87] where feasible to verify model fit improvements when assessing changes after consulting modification indices. The iterative process of model development included consideration of factor loadings.

#### 5.2.1. Financial Precarity (FP)

In constructing and evaluating the fits of the FP latent variable in partial fulfillment of Aim 1, the original model, assessed using EFA in which all items were allowed to load into both objective and subjective FP latent variables, produced a reasonable fit (KMO = 0.92; *χ*^2^ = 3822.4, *df* = 89, *p* < 0.001; *SRMR* = 0.049; *CFI* = 0.993; *TLI* = 0.991; *RMSEA* = 0.058, 90% CI [0.057, 0.06]). The final EFA model had variable C11_2012 (retirement account hardship withdrawal) removed due to 50% missingness and theoretical inappropriateness (retirement accounts are not ubiquitous). This final EFA model continued to have reasonable fit statistics (*χ*^2^ = 4879.999, *df* = 76, *p* < 0.001; *SRMR* = 0.059; *CFI* = 0.995; *TLI* = 0.994; *RMSEA* = 0.055, 90% CI [0.054, 0.057]). A review of EFA factor loadings (which ranged from -0.158 to 0.932 for objective FP and -0.312 to 0.934 for subjective FP) provided the basis for inclusion in the subsequent CFA model rendering. Items A40 (took on additional employment), J10 (large income drop), B4 (checking account overdraw), and D40 (use Medicaid / SNAP) were then excluded from the objective latent variable component of the FP measure, as they all had loadings below |0.40|. Item J10 (large income drop) loaded much more poorly onto the original objective latent variable component (0.029) than the subjective component (0.603) and was subsequently transferred to the latter, as the perception of an experience often matters more than the technical change [1]. The initial CFA fit for the FP measure was adequate (*χ*^2^ = 6201.99, *df* = 53, *p* < 0.001; *SRMR* = 0.057; *CFI* = 0.994; *TLI* = 0.993; *RMSEA* = 0.072, 90% CI [0.071, 0.074]). Suggested correlation of errors according to the modification indices that were theoretically supportable (based upon being related to FWB; [71], having cash available, and debt) were incorporated into the model (see Table 4), leading to improved fit (*χ*^2^ = 4083.276, *df* = 49, *p* < 0.001; *SRMR* = 0.048; *CFI* = 0.996; *TLI* = 0.995; *RMSEA* = 0.061, 90% CI [0.059, 0.062]). 

This final CFA model provided the basis for the FP component of the initial FBH model. Standardized CFA factor loadings ranged from 0.669 to 0.906 for objective FP and 0.592 to 0.875 for subjective FP. It should be noted that the objective and subjective latent variables provided a better fitting model on their own as correlated constructs (β = 0.872, *p* < 0.001), than modeling them as primaries to a second-order latent variable for FP. As such, they are referred to together as the measure of FP, albeit being distinct.

#### 5.2.2. Financial Self-Efficacy (FP)

A FSE latent variable was also constructed from NFCS items to approximate previously established measures [11,32]. The original model, assessed using EFA in which all items were allowed to load into both objective and subjective FP latent variables, produced a poor fit (KMO = 0.72; *χ*^2^ = 3057.167, *df* = 5, *p* < 0.001; *SRMR* = 0.081; *CFI* = 0.955; *TLI* = 0.91; *RMSEA* = 0.157, 90% CI [0.152, 0.161]). The final EFA model had variable J33_1 (worrying about running out of money in retirement) removed, despite an acceptable factor loading (0.412, *SE* = 0.006, *p* < 0.001), due to a high residual (0.41) and conflicting support for inclusion; Lown [32] includes this variable, but Hoge et al. [11] do not. The fit improved significantly, although some statistics were still not optimal, as there were only four remaining items included in the model (*χ*^2^ = 449.621, *df* = 2, *p* < 0.001; *SRMR* = 0.038; *CFI* = 0.992; *TLI* = 0.976; *RMSEA* = 0.094, 90% CI [0.087, 0.102]). The factor loadings for the remaining items ranged from acceptable (0.535) to strong (0.788) and were carried over into the CFA rendering. With no changes, the initial CFA fit for the FSE measure was identical to the final EFA rendering. Incorporated correlation of errors according to the modification indices that were theoretically supportable (based upon [M4] increased levels of financial knowledge contributing to [M1_1] higher confidence in carrying out money management tasks [89]) lead to improved, though still moderately poor fit (*χ*^2^ = 159.7976, *df* = 1, *p* < 0.001; *SRMR* = 0.021; *CFI* = 0.997; *TLI* = 0.983; *RMSEA* = 0.079, 90% CI [0.069, 0.09]). The comparatively large correlated error (−2.031) between the level of financial knowledge and confidence with money management could potentially be explained by the absence of a measure of one’s ability to navigate financial problems, a key feature of the established FSE scales [11,32]. Standardized CFA factor loadings ranged from 0.473 to 0.949 for FSE. This final CFA model (see Table 4) provided the basis for the FSE component of the initial FBH model.

#### 5.2.3. Financial Behavioral Health (FBH)

To fulfill Aim 2 and explore the relationship of the various components to FBH, the full FBH second-order latent variable model was constructed from the two FP components (objective and subjective first-order latent variables), the FSE latent variable, and the proxy item measure of FWB (J1, financial satisfaction). The original FBH model was assessed as hypothesized with mental health-related measures J33_40 (financial anxiety) and J33_41 (financial stress) as items regressed onto the FBH latent variable and without correlated errors. In this case, CFA rendering on the model produced a poor fit (*χ*^2^ = 34,696.4, *df* = 149, *p* < 0.001; *SRMR* = 0.08; *CFI* = 0.983; *TLI* = 0.981; *RMSEA* = 0.11, 90% CI [0.109, 0.111]). The model was then assessed as hypothesized with correlations among all principal components (FBH, FP, FSE, and FWB); the model did not converge. Consulting suggested modification indices, items J33_40 (financial anxiety) and J33_41 (financial stress), were correlated together and loaded onto the subjective FP latent variable based upon their perceived mental health responses to financial strain [1,51,90]. Correlations between individual items and constructs as determined in the prior EFA and CFA renderings were then incorporated, along with additional correlations between M1_2 (math ability) and M4 (financial knowledge; [91]), and M1_1 (confidence in money tasks) and M1_2 (math ability; [92]). The final CFA model produced a reasonably good fit (*χ*^2^ = 10,311.06, *df* = 140, *p* < 0.001; *SRMR* = 0.049; *CFI* = 0.995; *TLI* = 0.994; *RMSEA* = 0.061, 90% CI [0.06, 0.062]). The standardized factor loadings for FP component items of the model were acceptable (0.575) to strong (0.889). Although the standardized loadings of the FSE construct were reasonable (0.322 to 0.851), the unstandardized loadings were quite low: 0.043 to 0.115. By correlating errors among M1_2 (math ability), M4 (financial knowledge), and M1_1 (financial task ability), the model fit better, but this greatly lowered the factor loadings (unstandardized, without correlations, ranged from 0.25 to 0.51). Here, the decision was made to favor improved model fit over the strength of path loadings [75]. Latent variable loading onto FBH was also strong (from −0.78 to 0.991), with the FP constructs loading negatively (as precarity decreases, behavioral health improves). See Table 5 for complete standardized estimates.

### 5.3. FBH Model Analysis

To assess differences in FBH as it is presented differently by racial group affinity (Aim 3), measurement invariance tests (full and partial) were conducted. The FBH model was then analyzed by racial group, running the model separately for each to obtain distinct model fit statistics and estimates.

#### 5.3.1. Measurement Invariance

A test of model variance was conducted to determine if the application of the FBH model was different for the Black respondents as compared with the White respondents. The fit values are presented in Table 3. The same FBH model was run, with the same analysis parameters as the prior CFAs, with the addition of the group function to stratify the analysis (configural invariance). This base model fit well, similarly to the full sample model: (*χ*^2^ = 12,252.39, *df* = 280, *p* < 0.001; *SRMR* = 0.049; *CFI* = 0.994; *TLI* = 0.993; *RMSEA* = 0.067, 90% CI [0.066, 0.068]). To assess weak invariance, a constraint on factor loadings across both groups was applied. The model did not converge. Next, the constraint on factor loadings was released, and instead, a constraint on intercepts across both groups was applied. This produced a statistically significant change in the chi-squared difference test (Δ*χ*^2^ = 543.58, Δ*df* = 57, *p* < 0.001), suggesting weak invariance was not supported by the data. This was confirmed by adding a constraint on residual covariances to the constraint on intercepts, providing a stronger invariance rendering. An additional statistically significant change in the chi-squared difference test was produced (Δ*χ*^2^ = 63.35, Δ*df* = 8, *p* < 0.001). Next, a partial invariance test was conducted, in which factor loadings were constrained sequentially on every combination of latent variables (i.e., objective precarity only; subjective precarity only; objective + subjective precarity; self-efficacy only; objective + subjective precarity + self-efficacy; etc.). In no case did the model converge. As such, the FBH model may be concluded to be variant across racial groups. This indicates that the model is not conducive to cross-group usage and that racial group membership moderates the relationships represented by the FBH parameters [75].

#### 5.3.2. FBH Model by Racial Group

To better understand how the FBH model fit the data for each racial group differently, the FBH model was run separately for each sub-sample group. Model fit statistics are presented in Table 3. Whereas the model poorly fit data from the Black respondents (*χ*^2^ = 5851.964, *df* = 140, *p* < 0.001; *SRMR* = 0.127; *CFI* = 0.952; *TLI* = 0.941; *RMSEA* = 0.141, 90% CI [0.138, 0.145]), it had a good fit to data from the White respondents (*χ*^2^ = 6400.366, *df* = 140, *p* < 0.001; *SRMR* = 0.04; *CFI* = 0.997; *TLI* = 0.996; *RMSEA* = 0.051, 90% CI [0.05, 0.052]). Standardized factor loadings, as shown in Table 5, for Black respondents were lower across all objective FP items compared with loadings for White respondents, and were roughly comparable across subjective FP items. The loadings for FSE for Black respondents were all higher than or similar to those for White respondents, but were notably non-significant at the *p* < 0.05 level [93]. The White respondent data estimates for FSE were all statistically significant. Loadings onto the second-order FBH latent variable were strong and statistically significant when data from the White respondents were used. However, subjective FP had a loading of only −0.313 with Black respondents’ data. Additionally, the loading for FSE with Black respondents’ data was not statistically significant. Standardized estimates among the correlated errors were generally comparable between the two groups, except for the following. The correlation between the objective and subjective FP factors was notably stronger for Black respondents (β = 0.962) than for White respondents (β = 0.526). Also, correlated error estimates between J4 (difficulty paying bills) and J5 (rainy day savings) were notably stronger for Black respondents (−0.77) than for White respondents (−0.389).

### 5.4. FBH Model Applied to Investment Risk Willingness

The final analysis of the study was conducted to determine how FBH influences willingness to take investment risk, as potentially moderated by race (Aim 4). The measurement invariance test concluded that the racial group moderated the relationships represented within the FBH model. As such, further analysis of the racial group as a moderating variable in the applied model would not be appropriate, as model invariance should be established a priori to the application of the model. A general SEM analysis was thus used, regressing J2 (investment risk willingness) on the full FBH model and its correlated errors. This preliminary model did not converge. Control variables, including the constructed racial group, were brought into the model. An adequate model fit to the data was obtained (*χ*^2^ = 20,746.71, *df* = 230, *p* < 0.001; *SRMR* = 0.061; *CFI* = 0.983; *TLI* = 0.986; *RMSEA* = 0.068, 90% CI [0.068, 0.069]). The final path model is represented in Figure 3, with loadings, regressions, and correlations shown in Table 5. All factor loadings were moderate to strong (0.508 to 0.963), except the loading of M4 (financial knowledge) onto FSE (0.278), which could be explained by the incorporation of correlated errors, as already discussed. Factor loadings were all statistically significant. All controlled predictors of FBH and investment risk were supported by the literature, except for racial groups. Identifying as a Black respondent (compared with White) had no statistically significant relationship predicting FBH (β = −0.005, *p* = 0.428) and only a very small predictive impact on investment risk (β = −0.084, *p* < 0.001). Finally, it was found that when FBH increases by one standard deviation, one’s willingness to take investment risks increases by 0.256 standard deviations (*p* < 0.001). Comparatively, FBH had slightly more impact on risk willingness than being female (compared with male; β = −0.213, *p* < 0.001), and almost as much as increases in age (β = −0.269, *p* < 0.001). The *R*^2^ for the structural model was 0.368; the FBH model accounted for 37% of the variance in risk willingness, a relatively substantial effect size [94].

## 6. Discussion

This study explored FBH, including component indicators and measures that best reflect FBH. Additionally, model fit by racial group and the association between FBH and investment risk willingness were examined. Overall, the study provides empirical support for FBH, highlights its relevance to risk willingness, and emphasizes the importance of racial considerations when examining FBH and investment related constructs. Below, each aim of the study is further discussed.

Aim 1 of the study was to construct measures of FBH, including FP, to assess material and psychological well-being as impacted by negative financial conditions and perceptions. EFA and CFA were conducted using the full data set. Following the elimination of theoretically or statistically inappropriate items, this resulted in strong loadings for both the objective and subjective components of FP. It was found that FP is best modeled with the objective and subjective components as correlated factors, without FP itself as a second-order latent variable. Through the CFA process, the objective and subjective components were shown to be distinct, yet strongly related, as theoretically expected. The framing of FP with both material and psychological dimensions provides a more comprehensive measure than is typically used. In prior research, focus was on objective [15] or subjective conditions [14]; arguably, nuance is missing from these simpler FP measures. 

In addition to constructing a measure of FP, a proxy measure of FSE was developed from the available respondent questions. Its adequacy was modest, likely due to too few items and not enough of the right items (such as the ability to navigate difficulties) to fully capture the concept of self-efficacy [33]. Given that four items were retained to inform the latent variable, future research would be better served by using established FSE measures with more items to support them (i.e., [11,32]).

Aim 2 of the study explored the relationships the FBH constructs had on FBH; FP, FSE, and FWB together formed FBH as a second-order latent variable. The analysis suggests that FP is a strong and significant contributor to FBH, with a negative relationship. Thus, as the precariousness of one’s financial situation (in material and perceived terms) decreases, the sense of FBH improves. FSE did contribute to the model; however, more robust factor loadings would be expected with more appropriate variables. The proxy for FWB is also an important and statistically significant contributor to FBH. FWB has some conceptual similarities with subjective FP. Even though cross-loadings were not permitted in this model, to fully measure both components would require additional analysis to ensure the distinction of concepts. 

Additionally, FBH was ultimately modeled with financial anxiety and stress as subjective FP items, following the CFA construction of the component latent variables. If this revised modeling is retained in future research, one would expect a strong correlation with general mental health items (i.e., anxiety, stress, and depression), as there could be significant overlap in measured concepts. 

For Aim 3, differences were assessed in the FBH model fit according to racial group. Through the measurement invariance test, the FBH model was found to be variant; the measure presented differently for the two different racial groups. The overall FBH model fit the data from the White respondents very well, whereas it had poor model fit for the data from the Black respondents. A measure of partial invariance was unable to yield insight into any specific part of the model that was invariant, suggesting the whole model requires further refinement. However, further analysis yielded that between the Black and White respondents, there were notable differences in loadings in objective FP (all loadings for the Black respondents were lower) and FSE constructs (all loadings for the Black respondents were statistically not significant, despite financial knowledge loading more strongly than for the White respondents). 

A much higher factor correlation was measured between objective and subjective FP with data from the Black respondents than the White respondents. There was no evidence of divergent validity for the Black respondents, whereas it was much more evident for the White respondents [74]. This suggests that the differences between the objective and subjective aspects of FP are better captured for the White respondents and that the model is missing an important differentiator for the Black respondents. Similarly, there was a notable difference in correlated errors between having money set aside and difficulty paying bills (Black respondents, β = −0.77; White respondents, β = −0.389). This suggests that, for the Black respondents, there is something other than the objective latent variable that is highly influential in the relationship [74]. If the expectation to help others is a regular feature of one’s financial life, this could explain the extra correlation difference. 

The higher levels of collinearity for the Black respondents’ data warrant deeper consideration of the assumptions and philosophical orientation underpinning the survey structure and questions. Questions asking for the number of children who are financially dependent on the respondent or if the respondent has set aside money for their children’s college education are focused on the nuclear conception of the family. These types of questions neglect the sharing of food, occasional shelter, in-kind goods, and other assistance provided to extended kin or “as kin” friends. This sharing may have a significant impact on one’s financial state [45,47], yet it goes uncaptured due to the narrow question focus. Other surveys are starting to gather more collectivist-oriented data, such as the provision of unanticipated financial support to someone or expectations of repayment for money given [95] and remittance activity [96]. However, more research and data are needed to support generalizable findings. For the purpose of analyzing the influence of racial groups in Aims 3 and 4, the presence of bias in the questions and thereby the model makes it difficult to determine if scarcity-based racism influences FBH or the application of investment risk willingness.

Other collectivist groups may show similar model variance—Hispanic and Latinx, some Asian groups, people indigenous to North America, many immigrant populations (who may or may not have been included in the Black respondent group), people with lower income levels—some of whom are represented in the data set. It would be helpful to then assess the measurement invariance for these groups as well and compare the findings to the initial outcomes. More broadly, this is good research practice; the design of studies, surveys, models for analysis, and interventions all should account for cross-cultural differences that may impact validity [97,98,99]. Until then, and because the FBH model only fits well with data from the White respondents, the model should not be used for other groups until questions that better reflect Black and other groups’ collectivist experiences are incorporated. 

Furthermore, it is important to consider that if improving financial state and FBH is best completed according to relevant cultural contexts, then expecting everyone to use the same rules and tools will advantage those who operate according to the (individualist-oriented) contexts out of which the rules and tools were designed. Those from different (collectivist-oriented) cultural contexts may have trouble using or abiding by individualist-oriented tools and rules, ultimately finding FWB and security more difficult to attain. For those that do adopt the individualist-oriented approaches to wealth building, economic mobility can entail sacrificing bonds of financial interdependence [47]. 

The purpose of Aim 4 was to determine how FBH influences willingness to take investment risk, as potentially moderated by race. Overall, the applied FBH model analysis showed that much of one’s FBH is generated by lower FP, increased FSE, and FWB. This enhanced FBH leads to an increased willingness to take on investment risk. Having a degree of financial stability and capability would presumably put one in a position to take calculated investment risk. Within this relationship, scarcity may only have a small influence (possibly accounted for within the subjective FP construct), as well as be mitigated by other factors. As such, the expectation that scarcity would impact risk tolerance is difficult to assess and would require additional modeling components to distinguish it from subjective FP (if such a distinction can be made).

An applied assessment of racial group differences in scarcity’s impact on investment risk could not be conducted. As reported, the FBH model itself is moderated by racial group affiliation. Because the FBH model itself is variant between populations, assessing the impact of the race variable as a moderator on willingness to take investment risk (and influencing the effects of racism-based scarcity) cannot be determined; initial FBH measurement invariance should be established a priori. It should, however, be noted that when included as a control variable in the applied SEM model, change in the racial group (from White respondents to Black) had a very small contribution to risk willingness and no significant impact on FBH, contrary to what would be expected [50,100]. In this case, the design of the model and study likely have a strong overall influence that outweighs the otherwise well-documented relationship between racial identity and financial outcomes that is caused by structural racism and intermediated bias (interpersonal bias perceived by the individual). 

It was, however, observed that, as shown in Appendix B, Table A1 the Black respondents had a higher level of investment risk willingness than the White respondents. This difference was then eliminated in the final regression, which accounted for age, income, and gender, all known influences on risk willingness [50]. Interestingly, compared with the White respondents, Black respondents also had lower financial satisfaction, and higher financial anxiety and stress levels, all of which have been observed with lower risk willingness, not higher [50,52,65]. Willingness to take on greater or the same amount of risk when controlling for other known influencing factors could signal a shift from findings (or analytic approaches) [68] in earlier decades when White Americans were observed to have higher risk tolerance [49,50,67]. 

Gutter and Fontes [101] found initial support that differences between Black households and White households in assets owned in 2004 were more due to differential access to investment markets (a feature of bias and structural racism) and less a matter of risk tolerance. Although Black Americans’ ability to invest has historically been hampered by a lack of means, discrimination and distrust, it is not due to a lack of willingness [102]. In many cases, such as the 1921 destruction of “Black Wall Street” in Tulsa, Oklahoma [103], whole neighborhoods were razed by White vigilantes; Black Americans then lost their invested business and real estate wealth when insurance and city entities refused to compensate for losses. Such institutional racism thus inhibited future investment and for some families, may have perpetuated financial traumas that could manifest as decreased risk willingness [102]. However, others rebuilt, employing the spirit of collective resilience [103]. 

Today, the increasing prevalence and use of target date funds, the adoption of financial technology and robo-advisory services, and the use of investment apps (where the only requirements to entry are usually less money to invest than traditional accounts, access to the internet, and possibly a bank account) may play an additional factor in equalizing risk willingness for those who had traditionally been marginalized. These tools and investment vehicles rely heavily on algorithms to facilitate the risk allocation process, algorithms which are written by coders who ostensibly have financial independence in mind for clients. Thus, investing opportunity could be driving risk willingness rather than the other way around, in which Black investors accept the automated suggestions despite initial risk tolerance assessments at the point of enrollment. Additionally, more attention is being paid to improving the wealth-building capacity of Black Americans [104], and targeted financial education and inclusion efforts may also be having an impact on risk willingness [92]. Much more research is needed to understand the difference in risk willingness levels found in this study and then explore any actual change in trends and their causes.

Overall, the study found that FBH has a statistically significant and meaningful impact, comprising 37% of the variance in investment risk willingness. Although the contribution is relatively substantial, it should be appreciated in the broader context of what contributes to willingness to take investment risk, which could include gender, savings levels, income, opportunities to invest, and financial socialization [50]. Inasmuch as the gender wealth gap is due partly to structural factors but also to choices too [72,73,105], low FBH may contribute to wealth gaps as well. In total, this study prompts many additional questions and opens new areas for additional research.

### 6.1. Strengths and Limitations

This study used the large, nationally representative NFCS data set [69], which provided substantial power to handle the complex SEM modeling requirements. However, the representation of data from the notably smaller Black respondent sample precluded using multiple exclusive randomized subsamples for each exploratory and confirmatory analysis. Ideally, different samples of similar sizes would be used for each analysis to ensure replicability across samples [11,81] and to improve generalizability. In this case, a limitation of the study was that confirmation and application relied on the same data points. However, new constructs for FP and FBH were established upon sound theoretical and empirical support, and important distinctions for the appropriateness of the model’s use by racial group were explored. 

One notable challenge with using secondary data sets is the lack of control over available question items, their phrasing, and their measurement. In this case, the NFCS data set included only two mental health-related variables (financial anxiety and financial stress) and lacked other behavioral health items (related to generalized mental health and well-being conditions, substance use disorders, or physical well-being, for instance) or collectivist-oriented variables (such as financially related expectations to assist people beyond one’s own dependent children). Structural factors, such as employment opportunity, housing affordability, and explicit experiences with racism, which impact behavioral health, were likewise not included in the data. There is also a broad range of experiences and perspectives among people who categorized themselves as Black or African American in the data set (including, for instance, first- or second-generation immigrants.) Having a broader selection of demographic categories, especially in terms of culture and ethnicity or regional differences, would allow for a better assessment of differences and common traits. Additionally, certain measures, such as income volatility, are best measured repeatedly over time (ideally, monthly) [46] in order to capture episodic impacts on mental health. Furthermore, certain concepts require more questions to be adequately captured, such as the seven-item Financial Anxiety Scale [106] or the 10-item Scale of Economic Self-Efficacy [11]. If sponsors of large data collection undertakings are interested in understanding the connections between racism, wealth, and well-being, then the relevant indicators need to be collected. In the absence of key theoretical components (such as the ability to navigate difficult financial situations as a part of FSE), a bi-factor exploratory SEM (ESEM) [107] approach may be useful in order to handle the large amounts of correlated residual errors.

### 6.2. Future Research and Implications

In constructing, analyzing, and applying the FBH model, several implications for future research warrant consideration. The dual-faceted FP construct included elements of the CFPB’s Financial Well-Being Scale [71] in the subjective component. Because FP then measures aspects of well-being in a more holistic manner, this FP arrangement could be a better measure of financial state than the Financial Well-Being Scale alone. If so, the full FBH model may need to be altered to fully integrate FWB concepts into the subjective component of FP, eliminating it as a distinctly third component of the FBH model. Furthermore, considering that the FBH model is known to be variant by racial group, it would also be important to conduct racial group measurement invariance studies either independently on the Financial Well-Being Scale itself or at least to reassess invariance when it is more fully integrated into the FBH model. 

The next steps for research on the FBH model include more deeply exploring the relationships between a more robust version of FSE, FP, and, if still retained, FWB. Additionally, more contextually (collectivist) representative questions, measuring invariance across other populations, and then refining the scale and testing the FBH measure are needed. Including the collectivist or sharing-oriented questions that may help address the measure variance issue, as many people share, but in different ways. Other invariant questions may be found to be more appropriate for each of the measured constructs. Establishing that the present set is collectively variant aids with this process by signaling those that may be unsuitable. As a part of this model testing, qualitative research approaches should be used to ensure various collectivist cultural perspectives on money management and financial responsibilities are adequately captured. Additionally, profiles of populations (with a greater breakdown between African Americans, Black Latinx populations, and immigrants of African descent) exhibiting high and low FBH will need to be generated to determine key characteristics and factors that contribute to resilience or vulnerability and under what conditions.

Once the FBH model is found to be adequately reliable and valid, its relationship as a contributor to the elements of behavioral health (mental, physical, coping, and social health) should be thoroughly assessed. FBH may also have implications for other micro-level concerns, such as family and child well-being; researchers have already begun to explore similar topics, such as the impact of income volatility on family food insufficiency [108], divorce [109], and fertility [110]. Various social determinants of health may also have a relationship with FBH, such as job and housing stability, social service spending, and use of employer-sponsored benefits. Exploring racism as a determinant, which influences other determinants [111], and thus its impact on FBH, should be an integral part of this research.

In the financial domain, FBH could impact the odds of investing in more risky or conservative financial products, and ultimately have an impact on levels of wealth. Two particular studies found that those with higher levels of anxiety were more averse to investment risk [52,65]. With a nuanced understanding of FBH and how it impacts and reflects the experiences of different people, more inclusive financial products and services could be designed to meet divergent needs. Additionally, as Colman [67] found, there may be relatively little difference between White and Black investors’ risk willingness (after controlling for household wealth levels). Rather, differences may manifest in the choice of products, which could be where individualist versus collectivist values become more relevant. Coleman’s data found that White investors held the greatest amount of their assets in stocks (23.6%), followed by retirement funds (14.9%). Black investors however, held the greatest amount of their assets in life insurance (22.4%), followed by pensions (21%). Differences in the cultural importance of long-term financial security as a legacy for family members could explain this difference, but confirming this, along with incorporating explicitly collectivist-oriented questions, is needed.

Finally, in his chapter on financial risk tolerance, Grable [50] highlighted several consumer finance research topics in need of further exploration, including factors that influence financial risk-taking behavior, whether cultural background affects risk tolerance, and how emotional states contribute to risk tolerance. The present study on FBH begins to offer some insight into these topics and helps formulate some clear steps for future research. 

This study provided the initial steps for the creation of a FP measure and a FBH scale. Although the reliability and validity of both measures need to be assessed, social workers and behavioral health practitioners can use vetted measures such as the Financial Well-Being Scale [71], the Scale of Economic Self-Efficacy [11], the Financial Anxiety Scale [106], and other measures and questions to learn the financial perceptions and experiences of clients. Recent literature helps practitioners recognize the connection financial state and perception have with other behavioral health issues and that outcomes may be inter-influential [112]. When warranted, practitioners should work collaboratively with clients to establish values-oriented and culturally informed budgets and good money management behaviors, especially if there is a cross-cultural relationship between the practitioner and client. Differences in approaches and presumed contexts could influence what gets prescribed and the subsequent success or failure of outcomes. Likewise, financial advisors and counselors helping people with money management should appreciate the implications of behavioral health on financial concerns.

Although the present study found support for similar investment risk willingness between Black and White respondents, it should not be presumed that actual investing activity will subsequently achieve equal outcomes. Encouraging people to invest is important, but so is creating an environment of equal investment opportunities. Disparities in wealth accumulation will persist until legislation, policies, and programs are expressly designed to address those disparities and are carried out by leadership intent on seeing change. 

By neglecting collectivist orientations to handling and thinking about money, FINRA and other entities surveying personal financial topics are potentially missing a significant component of how people relate to their finances. This can hinder educational efforts and outcomes that seek to improve the financial security, wealth, and well-being of populations that have historically been marginalized. Culturally appropriate research questions and analysis into how these populations think about and treat money will then ensure a better design of financial policies, products, and services to meet their needs. Finance-oriented institutions need to gather more health data, and conversely, health-oriented institutions (including federal health agencies) need to collect more financial data. The Medical Expenditure Panel Survey (MEPS) [113] has some of this dual-focus data; however, some of it can be difficult for researchers to fully access. Beyond data gathering, benefit programs and policies designed to help individuals and families financially should be coupled with targeted supportive policies and education that address FP and FSE. Such financially based programming could bolster a person’s well-being and, thereby, their ability to become self-sufficient and able to support others within their collective networks.

## 7. Conclusions

FBH has the potential to influence and be influenced by multiple other behavioral health domains and, as applied, has implications for willingness to take investment risk. In Aim 1 of the study, components of FBH—FP, FSE, and FWB—and their relationships with FBH were constructed and assessed. In Aim 2, the full FBH model was measured for overall fit. The study then explored differences in model fit according to racial affinity group, focusing on respondents who self-selected Black and/or White (Aim 3). Lastly, in Aim 4, the authors explored how FBH influences willingness to accept investment risk. 

Many connections have been made between distinct financial and behavioral health domains (such as between debt and depression), yet this is the first study to look comprehensively at the overall FBH construct and provide any theoretical or empirical treatment of its overarching conceptualization. In addition, the study established a more holistic measure of FP, comprising both objective and subjective considerations; prior conceptualizations of precarity in the financial domain focused exclusively on one area or the other. The finding of significant variance in the FBH measure between the two sub-group samples suggests that either key items are being differently interpreted or are missing from the data set—including questions that capture collectivist-oriented thoughts and patterns of behavior. Lastly, the study illustrated that FBH contributes roughly as much to investment risk willingness as do a person’s gender and age, two well-acknowledged influences. Broadly, the project expands conceptions of the financial domain beyond just a social determinant of health. It helps us appreciate the psycho-behavioral impact finances have on people’s well-being and decisions and how this may ultimately influence the propensity to invest or not—thereby perpetuating wealth gaps.

## Figures and Tables

**Figure 1 ijerph-20-05835-f001:**
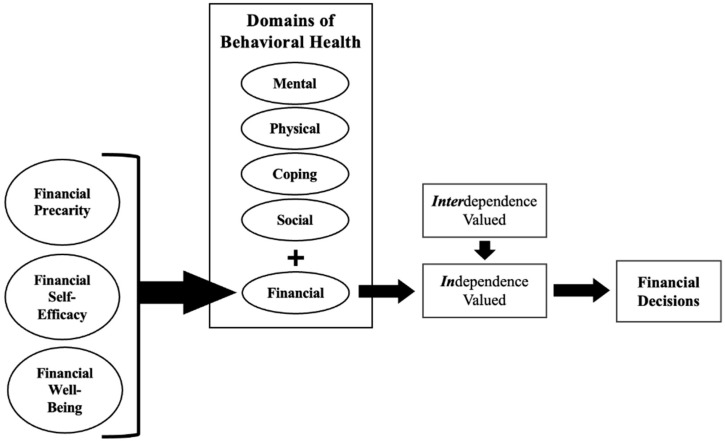
Conceptual model of financial behavioral health (FBH).

**Figure 2 ijerph-20-05835-f002:**
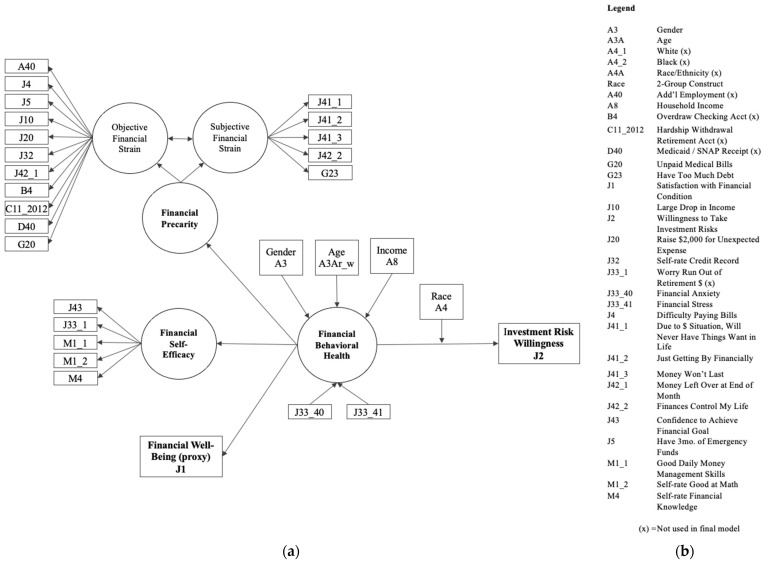
(**a**) Hypothesized path model of FBH with impact on investment risk willingness; (**b**) Legend of total variable items used in study.

**Figure 3 ijerph-20-05835-f003:**
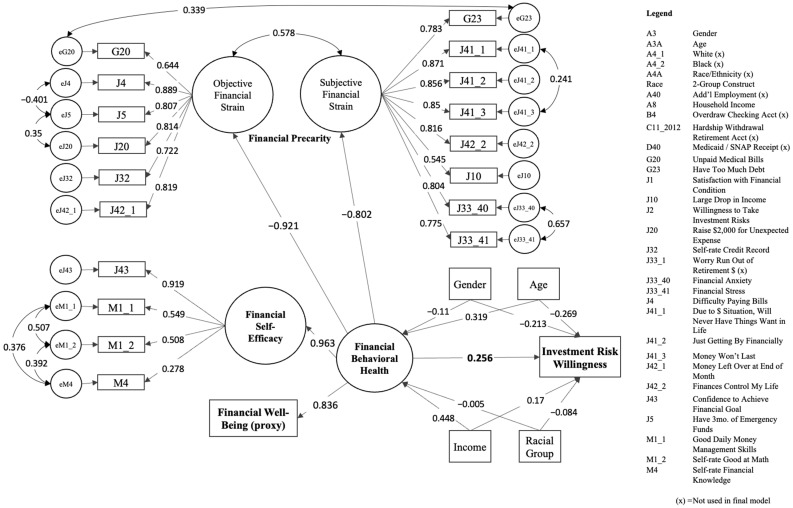
Final path diagram of FBH model applied to investment risk willingness.

**Table 1 ijerph-20-05835-t001:** Items for Financial Precarity (FP) and Financial Self-Efficacy (FSE) Scales.

Measure	Name	Number (Position)	Scale Score	Reverse Coded	Informed by CFPB FWB Scale
Financial Precarity					
Objective Measures	Took on additional employment	A40 (22)	1 = Yes, 2 = No	R	
	Difficulty paying bills	J4 (32)	1 = Very difficult, 2 = Somewhat difficult, 3 = Not at all difficult	R	
	Having three months’ worth of emergency funds	J5 (34)	1 = Yes, 2 = No		
	Experienced a large income drop	J10 (38)	1 = Yes, 2 = No	R	
	Confidence to handle a $2000 financial shock	J20 (39)	1 = I am certain I could come up with the full $2000, 2 = I could probably come up with $2000, 3 = I could probably not come up with $2000, 4 = I am certain I could not come up with $2000		Yes
	Self-rating of credit	J32 (40)	1 = Very bad, 2 = Bad, 3 = About average, 4 = Good, 5 = Very good	R	
	Frequency of having money left over at the end of the month	J42_1 (47)	1 = Never, 2 = Rarely, 3 = Sometimes, 4 = Often, 5 = Always	R	Yes
	Overdrawing on a checking account	B4 (52)	1 = Yes, 2 = No	R	
	Hardship withdrawal from retirement account	C11_2012 (66)	1 = Yes, 2 = No	R	
	Use of Medicaid/SNAP	D40 (70)	1 = Yes, 2 = No	R	
	Outstanding medical debt	G20 (85)	1 = Yes, 2 = No	R	
Subjective Measures	Because of my financial situation, I feel like I will never have the things I want in life	J41_1 (44)	1 = Does not describe me at all, 2 = Describes me very little, 3 = Describes me somewhat, 4 = Describes me very well, 5 = Describes me completely		Yes
	I am just getting by financially	J41_2 (45)	1 = Does not describe me at all, 2 = Describes me very little, 3 = Describes me somewhat, 4 = Describes me very well, 5 = Describes me completely		Yes
	Concerned that money I have or will save won’t last	J41_3 (46)	1 = Does not describe me at all, 2 = Describes me very little, 3 = Describes me somewhat, 4 = Describes me very well, 5 = Describes me completely		Yes
	My finances control my life	J42_2 (48)	1 = Never, 2 = Rarely, 3 = Sometimes, 4 = Often, 5 = Always		Yes
	I have too much debt right now	G23 (104)	1 = Strongly disagree, 2, 3, 4 = Neither agree nor disagree, 5, 6, 7 = Strongly agree		
Financial Self-Efficacy					
	Confidence in ability to achieve a financial goal	J43 (49)	1 = Not at all confident, 2 = Not very confident, 3 = Somewhat confident, 4 =Very confident		
	Worry about running out of money in retirement	J33_1 (41)	1 = Strongly disagree, 2, 3, 4 = Neither agree nor disagree, 5, 6, 7 = Strongly agree	R	
	Good at dealing with day-to-day financial matters, such as checking accounts, credit and debit cards, and tracking expenses;	M1_1 (109)	1 = Strongly disagree, 2, 3, 4 = Neither agree nor disagree, 5, 6, 7 = Strongly agree		
	Self-assessed ability, being pretty good at math	M1_2 (110)	1 = Strongly disagree, 2, 3, 4 = Neither agree nor disagree, 5, 6, 7 = Strongly agree		
	Self-assessed overall financial knowledge.	M4 (111)	1 =Very low, 2, 3, 4, 5, 6, 7 = Very high		

For each scale, respondents also had options to indicate “don’t know” (98) or “prefer not to say” (99). Both were recoded as missing (NA). Measures were reverse coded so that a higher FP score represented more precarity. A higher FSE score indicated greater levels of confidence and ability.

**Table 2 ijerph-20-05835-t002:** Demographic characteristics of respondents.

Demographic Characteristics	Full Sample	Combined Sub-Sample	White Respondent Sub-Sample	Black Respondent Sub-Sample
	(*N* = 27,091)	(*n* = 24,124)	(*n* = 21,289)	(*n* = 2835)
	M (SD)/# (%)	M (SD)/# (%)	M (SD)/# (%)	M (SD)/# (%)
Age	48 (17)	49 (17)	50 (17)	41 (15)
Gender				
Male	11,956 (44%)	10,680 (44%)	9427 (44%)	1253 (44%)
Female	15,135 (56%)	13,444 (56%)	11,862 (56%)	1582 (56%)
Race/Ethnicity				
White non-Hispanic	20,099 (74%)	20,099 (83%)	20,099 (94%)	0 (0%)
Black non-Hispanic	2576 (10%)	2576 (11%)	0 (0%)	2576 (91%)
Hispanic (any race)	2338 (9%)	922 (4%)	814 (4%)	108 (4%)
Asian non-Hispanic	1210 (4%)	0 (0%)	0 (0%)	0 (0%)
Other, Multiple non-Hispanic Ethnicities	868 (3%)	527 (2%)	376 (2%)	151 (5%)
Edu Level Attained				
<HS	697 (3%)	609 (3%)	522 (2%)	87 (3%)
High school—Diploma	4900 (18%)	4452 (18%)	3914 (18%)	538 (19%)
High school—GED/Alt	1919 (7%)	1699 (7%)	1480 (7%)	219 (8%)
Some college	7263 (27%)	6531 (27%)	5565 (26%)	966 (34%)
Associate’s Degree	2864 (11%)	2528 (10%)	2220 (10%)	308 (11%)
Bachelor’s Degree	5905 (22%)	5192 (22%)	4700 (22%)	492 (17%)
Post graduate degree	3543 (13%)	3113 (13%)	2888 (14%)	225 (8%)
Household Income				
<$15,000	3041 (11%)	2632 (11%)	2053 (10%)	579 (20%)
$15,000–$24,999	2804 (10%)	2509 (10%)	2139 (10%)	370 (13%)
$25,000–$34,999	2934 (11%)	2590 (11%)	2246 (11%)	344 (12%)
$35,000–$49,999	3917 (14%)	3497 (14%)	3099 (15%)	398 (14%)
$50,000–$74,999	5259 (19%)	4694 (19%)	4198 (20%)	496 (17%)
$75,000–$99,999	3856 (14%)	3472 (14%)	3106 (15%)	366 (13%)
$100,000–$149,999	3439 (13%)	3099 (13%)	2901 (14%)	198 (7%)
≥$150,000	1841 (7%)	1631 (7%)	1547 (7%)	84 (3%)
No. of Financially Dependent Children				
Do not have any children	9006 (33%)	7823 (32%)	6832 (32%)	991 (35%)
No financially dependent children	8471 (31%)	8004 (33%)	7438 (35%)	566 (20%)
1 financially dependent child	4198 (15%)	3635 (15%)	3080 (14%)	555 (20%)
2 financially dependent children	3255 (12%)	2789 (12%)	2418 (11%)	371 (13%)
3 financially dependent children	1355 (5%)	1186 (5%)	964 (5%)	222 (8%)
4 or more financially dependent children	806 (3%)	687 (3%)	557 (3%)	130 (5%)
Financial Satisfaction	5.72 (2.88)	5.74 (2.89)	5.79 (2.84)	5.31 (3.16)
Financial Anxiety	4.51 (2.02)	4.49 (2.02)	4.46 (2.02)	4.73 (2.06)
Financial Stress	4.12 (2.07)	4.09 (2.08)	4.05 (2.07)	4.35 (2.16)
Investment Risk Willingness	4.91 (2.68)	4.85 (2.68)	4.79 (2.61)	5.36 (3.07)

Financial satisfaction was scaled 1 (not at all satisfied) to 10 (extremely satisfied). Financial anxiety and financial stress were scaled 1 (strongly disagree) to 7 (strongly agree). Investment risk willingness was scaled from 1 (not at all willing) to 10 (very willing). # (%) refers to count as number and percent of sample.; “*N*” is full sample, “*n*” is subsample.

**Table 3 ijerph-20-05835-t003:** Model fit indices and chi-square difference tests.

Model Construction	n	χ^2^ (*df*) ***	SRMR	CFI	TLI	RMSEA [90% CI]	Δχ^2^ (*df*)
FP							
Original EFA	12,288	3822.4 (89)	0.049	0.993	0.991	0.058 [0.057, 0.06]	
Final EFA	20,590	4879.999 (76)	0.059	0.995	0.994	0.055 [0.054, 0.057]	(NA—different variables)
Original CFA	22,302	6201.99 (53)	0.057	0.994	0.993	0.072 [0.071, 0.074]	8582.2 (76) ***
Final CFA	22,302	4083.276 (49)	0.048	0.996	0.995	0.061 [0.059, 0.062]	3088.2 (4) ***
FSE							
Original EFA	24,849	3057.167 (5)	0.081	0.955	0.91	0.157 [0.152, 0.161]	
Final EFA	25,269	449.621 (2)	0.038	0.992	0.976	0.094 [0.087, 0.102]	(NA—different variables)
Original CFA	25,269	449.621 (2)	0.038	0.992	0.976	0.094 [0.087, 0.102]	0 (0)
Final CFA	25,269	159.7976 (1)	0.021	0.997	0.983	0.079 [0.069, 0.09]	643.42 (1) ***
FBH							
Original CFA	19,232	34,696.4 (149)	0.08	0.983	0.981	0.11 [0.109, 0.111]	
CFA w/all way correlations							DNC
Final CFA	19,232	10,311.06 (140)	0.049	0.995	0.994	0.061 [0.06, 0.062]	5297.2 (9) ***
Model Assessments							
Multiple Group Analysis	W = 17,192, B = 2040	12,252.39 (280)	0.049	0.994	0.993	0.067 [0.066, 0.068]	
MI: loadings constrained	W = 17,192, B = 2040						DNC
MI: intercepts constrained	W = 17,192, B = 2040	13,191.54 (337)	0.049	0.994	0.994	0.063 [0.062, 0.064]	543.58 (57) ***
MI: intercepts + residual covariances constrained	W = 17,192, B = 2040	13,367.8 (345)	0.049	0.994	0.994	0.063 [0.062, 0.064]	63.35 (8) ***
Partial Invariance (loadings constrained)							DNC for any combination of freed latent factors
Black respondent model fit	2040	5851.964 (140)	0.127	0.952	0.941	0.141 [0.138, 0.145]	
White respondent model fit	17,192	6400.366 (140)	0.04	0.997	0.996	0.051 [0.05, 0.052]	
Applied Model _(no covariates)_							DNC
Applied Model _(with covariates)_	19,031	20,746.71 (230)	0.061	0.983	0.986	0.068 [0.068, 0.069]	

*** *p* < 0.001. Δχ^2^ (*df*) was conducted using LRT, where feasible. NA = Not applicable. DNC = Did not converge. MI = Measurement Invariance.

**Table 4 ijerph-20-05835-t004:** Component CFA standardized factor loadings.

Component Models		Financial Precarity				Financial Self-Efficacy	
		Objective		Subjective			
Item No.	Description	Estimate	*p*	Estimate	*p*	Estimate	*p*
A40	Add’l Employment (x)						
J4	Bill Difficulty	0.906					
J5	Emergency Funds	0.812	***				
J10	Income Drop			0.592			
J20	Raise $2000	0.831	***				
J32	Self-Rate Credit	0.754	***				
J42_1	Money Left Over	0.786	***				
B4	Overdraw Checking (x)						
D40	Medicaid/SNAP (x)						
G20	Medical Debt	0.669	***				
J41_1	Never Have Wants			0.861	***		
J41_2	Just Getting By			0.875	***		
J41_3	Money Won’t Last			0.811	***		
J42_2	Controls My Life			0.801	***		
G23	Too Much Debt			0.751	***		
J43	Goal Confidence					0.473	
M1_1	Management Skills					0.949	***
M1_2	Good at Math					0.582	***
M4	Financial Knowledge					0.892	***
Covariances							
Obj ⬄	Subj	0.872	***				
J41_1 ⬄	J41_3	0.103	***				
J5 ⬄	J20	0.12	***				
J4 ⬄	J5	−0.118	***				
G20 ⬄	G23	0.151	***				
M1_1 ⬄	M4					−2.031	***

*** *p* < 0.001. Only final model factor loadings reported. All estimates are standardized values. An (x) indicates item was tested, but excluded, from final CFA model. ⬄ indicates correlated error between indicated items.

**Table 5 ijerph-20-05835-t005:** Full model CFA and SEM standardized factor loadings.

Latent Variables		CFA FBH Combined Groups		CFA FBH Black Sub-Group		CFA FBH White Sub-Group		SEM Investment Risk	
		*n* = 19,232		*n* = 2040		*n* = 17,192		*n* = 19,031	
Item No.	Description	Estimate	*p*	Estimate	*p*	Estimate	*p*	Estimate	*p*
	Obj ⇨								
J4	Bill Difficulty	0.889	***	0.786	***	0.904	***	0.889	***
J5	Emergency Funds	0.825	***	0.749	***	0.834	***	0.807	***
J20	Raise $2000	0.837	***	0.772	***	0.844	***	0.814	***
J32	Self-Rate Credit	0.756	***	0.632	***	0.77	***	0.722	***
J42_1	Money Left Over	0.808	***	0.758	***	0.818	***	0.819	***
G20	Medical Debt	0.649	***	0.435	***	0.674	***	0.644	***
	Sub ⇨								
J10	Income Drop	0.575	***	0.594	***	0.575	***	0.545	***
J41_1	Never Have Wants	0.876	***	0.86	***	0.879	***	0.871	***
J41_2	Just Getting By	0.872	***	0.809	***	0.877	***	0.856	***
J41_3	$ Won’t Last	0.849	***	0.845	***	0.851	***	0.85	***
J42_2	Controls My Life	0.812	***	0.763	***	0.819	***	0.816	***
G23	Too Much Debt	0.763	***	0.682	***	0.77	***	0.783	***
J33_40	Financial Anxiety	0.804	***	0.776	***	0.808	***	0.804	***
J33_41	Financial Stress	0.773	***	0.769	***	0.774	***	0.775	***
	FSE ⇨								
J43	Goal Confidence	0.851	***	0.865	0.101	0.847	***	0.919	***
M1_1	Management Skills	0.556	***	0.53	0.098	0.553	***	0.549	***
M1_2	Good at Math	0.322	***	0.306	0.098	0.326	***	0.508	***
M4	Financial Knowledge	0.518	***	0.603	0.1	0.517	***	0.278	***
	FBH ⇨								
Obj	Obj	−0.894	***	−0.887	***	−0.909	***	−0.921	***
Sub	Sub	−0.78	***	−0.313	***	−0.84	***	−0.802	***
FSE	FSE	0.991	***	0.987	0.104	0.99	***	0.963	***
J1	Financial Satisfaction	0.836	***	0.845	***	0.839	***	0.836	***
Covariances									
Obj ⬄	Subj	0.604	***	0.962	***	0.526	***	0.578	***
J41_1 ⬄	J41_3	0.237	***	0.235	***	0.234	***	0.241	***
J5 ⬄	J20	0.361	***	0.473	***	0.34	***	0.35	***
J4 ⬄	J5	−0.44	***	−0.77	***	−0.389	***	−0.401	***
G20 ⬄	G23	0.356	***	0.465	***	0.321	***	0.339	***
J33_40 ⬄	J33_41	0.666	***	0.614	***	0.672	***	0.657	***
M1_2 ⬄	M4	0.408	***	0.393	***	0.41	***	0.392	***
M1_1 ⬄	M4	0.378	***	0.313	***	0.388	***	0.376	***
M1_1 ⬄	M1_2	0.5	***	0.431	***	0.51	***	0.507	***
Regressions									
Inv Risk ⇦	Gender							−0.213	***
	Income							0.17	***
	Age							−0.269	***
	Racial Group							−0.084	***
FBH ⇦	Gender							−0.11	***
	Income							0.448	***
	Age							0.319	***
	Racial Group							−0.005	0.428
Inv Risk ⇦	FBH							0.256	***

*** *p* < 0.001. Only final model factor loadings reported. All estimates are standardized values. Obj = Objective FP. Subj = Subjective FP. ⇨ indicates loading on latent factor. ⬄ indicates correlation. ⇦ indicates regression.

## Data Availability

A disaggregated version of a publicly available data set was analyzed in this study. Publicly available data can be found here: https://www.finrafoundation.org/knowledge-we-gain-share/nfcs/data-and-downloads (accessed on 24 January 2023). The disaggregated version may be obtained upon agreement from the FINRA Investor Education Foundation.

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
