# Peer review of "Financial Behavioral Health and Investment Risk Willingness: Implications for the Racial Wealth Gap"

_ijerph, 2023, doi:10.3390/ijerph20105835_

Round 1

Reviewer 1 Report

The article is about Financial Behavioral Health. It is a very interesting topic. Authors carried out an extensive analysis of the investigated problem - processed results in a separate section. Conclusion of the article would be appropriate to elaborate in more detail given the scope of the paper. In the first phase, the contribution attracts attention, but gradually it becomes unclear. The article is very extensive.

Reviewer 2 Report

This study is well-organized and written. However, there are some problems with the research methodology. The research methodology should be improved.
1. The research framework diagram (figure 2) is significantly different from the statistical analysis diagram (figure 3). e.g., J33_40 & J33_41

2. There is no independent variable in Figure 2 & 3, which obviously does not conform to the basic specifications of SEM model. I know that is FBH, however, the authors' presentation contains some problems.

3. The residual correlation (in Figure 3) means that the residual is not independent. The residual correlation should not be used to improve the model's fitness. There are other ways to improve the model's fitness. For example, the ADF (WLS) method can be used if the sample size is larger than 2000.

4. What software used is not mentioned in the article for SEM analysis

5. The sub-topic is Social Wealth Gap which should conduct a group difference test (e.g., simple t-test) rather than on the model as a control variable.

Reviewer 3 Report

The topic of the article is important, but the article at its present form is too unpolished and confusing for the reader to be accepted for publication. The tables should be prepared to "stand alone", without unexplained abbreviations. Now many of the tables and figures are confusing and they also contain incorrect/strange abbreviations "A3A" etc.

The references-list should preferrably contain peer-reviewed work, not talks.

"Wolfsohn, R. Financial Behavior Health for You and Your Clients. Presented at the Center for Financial Social Work"

For instance, the beginning of the Discussion is full of abbreviations. The Discussion should start with a brief and easily understandable rewrite of the results. Now it is somewhat confusing for a naive reader.

"6. Discussion
Aim 1 of the study was to construct measures of FBH, including FP to assess material
and psychological well-being as impacted by negative financial conditions and percep-
tions. EFA and CFA were conducted using the full data set. Following the elimination of
theoretically or statistically inappropriate items, this resulted in strong loadings for both
the objective and subjective components of FP"

In general the article is too confusing for publication.

Reviewer 4 Report

You have embarked on the analysis of an exceedingly complex problem using equally complex modelling, and highly technical language.

Who is [Author] in line 55?

Have you contacted Wolfsohn (line 57) to ask how they derived their ideas?

Is there any research on the role of personality type into investment decisions, etc?

My area of expertise is not in the area of complex modeling and hence my comments are somewhat general.

Perhaps consider who is the audience for the paper (academics, policy makers, etc) and ensure that the intended audience understands the language and its implications.

I note there is a significant difference between average age for the white and black groups. It would probably be useful if you could stratify Table 2 by age group. Maybe place this in an Appendix with comments.

Indeed how was adjustment for age made in the model?

Lines 445-457 is there any literature covering the strengths and pitfalls of this modelling approach. How is it most commonly applied in the literature. 

Are there any hidden assumptions in the modeling?

Is there any way to use charts to illustrate the impact of some of the key variables.

I am going to suggest that you resubmit after a major revision with the aim to communicate what is a good study in a way which is more accessable.

For example, is the 'problem' open to resolution and how?

Reviewer 5 Report

The article is very interesting. It raises a topic that is not often described in the literature and in my opinion is extremely important. Behavioral finance, although not a new discipline, is rare. It seems to me that all attempts to study human behavior, regardless of their cross-section or even scope, are most desirable. Thus, I thank the authors for taking up this topic.

The article is well written both technically and methodologically. The introduction is short and to the point, without unnecessary informations. The literature review and methodological part is very well implemented. Everything is described in a way that is extremely understandable for the reader and reinforced with graphical presentations in the form of block diagrams. The Discussion part is extremely important to me. The authors devoted a very wide range of their article to discussion. In addition, they summarized the whole thing. The literature review contains over 100 items (110 to be exact).

Nothing more nothing less.

I congratulate the authors of both the idea and its implementation.

Good job :)

Reviewer 6 Report

Dear authors, congratulations on your article.

I think it would be convenient to add as a limitation the differences that exist between the two samples used, both in size and in the characteristics of the groups, which may affect the generalization of the study.

Reviewer 7 Report

Thank you for your very solid paper.

The theoretical background analysis is consistent.

The large sample size is a great plus, but being a secondary data set also brings some limitations, as you yourself pointed out in the limitations section.

The statistical analysis is advanced, complex and adequate for purpose. 

The implications of the findings are analyzed and interpreted correctly.

Reviewer 8 Report

The study is original.  The model used is clearly described and this allows for its application by other authors. The results are interpreted thoroughly. 

Reviewer 9 Report

Referee report on "Financial behavioral health and investment risk willingness: Implication for the racial wealth gap"

The authors study an important topic. The paper is well motivated and conceptualized. The introduction is nice and informative. It is written in a plain language, which is easy to follow. My comments are below. 

 Comments

1. To make your argumentation more balanced, in the end of the first paragraph on Page 6 (line 245) the authors could add that “Different investment strategies between Black Americans and White Americans can also result from preferences toward conspicuous consumption (Charles et al., 2009) or different levels of education (Azarnert, 2010).”

2. My next comment is more substantial. In view of the possible historical self-selection in Black migration from the South and the possible differences across the two regions in the United States, the paper will benefit from the comparison of Black residing in the South to Black residing in the North. This comparison could probably help the authors to strengthen their interpretation of the results.

This brings me to my last comment.

3. Although it is a plausible reason for the observed differences between Black Americans and White Americans, throughout the paper the authors provide no real evidence of the structural racism. It equally might be, for example, perceived racism, rather than structural racism. Moreover, the term “structural racism” could be interpreted as offensive for the whole American society. I would strongly recommend to the authors to avoid extensive use of this term and use more neutral and less offensive terms, as some of the authors cited in this study do.

 Minor (editorial) comment

4. Re-check and finalize your statements throughout the paper. For example, lines 199 – 203 and also line 55.

References

Charles, K. K., Hurst, E., & Roussanov, N. (2009). Conspicuous consumption and race. Quarterly Journal of Economics, 124, 425–467

Azarnert, L.V. (2010) Juvenile imprisonment and human capital accumulation. Journal of Economic Inequality, 8, 23–33

Round 2

Reviewer 2 Report

After reading the attached files, I have no comments for this paper and suggest acceptance. 

The authors have addressed the previous comments.

Reviewer 4 Report

Thank you for your well reasoned response to me questions. I am happy to say accept.

Well done.